# Structural insight into hierarchical DNMT3A autoinhibition and its dysregulation in disease

Jiuwei Lu [1], Emily Vig[2,5], Jianbin Chen[1,2,5], Kristjan H. Gretarsson [3,5], Nelli Khudaverdyan[1,2], Zengyu Shao [1,2], Chao Lu [3], Chia-en A. Chang [2,4] & Jikui Song [1,2] ✉

DNA methyltransferase DNMT3A-mediated DNA methylation is important for genomic imprinting and transcriptional regulation. However, how the regulatory domains of DNMT3A cooperate with its methyltransferase domain and histone marks to orchestrate genomic methylation remains unclear. Here we report the cryo-EM structure of DNMT3A2 with regulatory factor DNMT3L, revealing an intricate domain interaction underlying multilayered autoinhibition. The PWWP domain interacts with the ADD and methyltransferase domains to block the target recognition domain and the H3K36me2-binding pocket, thereby coupling the H3K36me2 binding with DNMT3A activation, adding a layer of allosteric regulation distinct from the previously characterized ADD-H3K4me0 regulation. Molecular dynamics simulations of the DNMT3A-DNMT3L complex further reveals that relief of DNMT3A autoinhibition involves disengagement of the CpG-recognition loop of the target recognition domain from autoinhibitory interaction, leading to enhanced accessibility of the target recognition domain loop for DNA binding and DNMT3A activation. Importantly, our combined structural, biochemical and genomic methylation analysis demonstrates that disrupting the PWWP-ADD interaction by disease-associated DNMT3A mutations leads to impaired DNMT3A autoinhibition and substrate specificity, providing a potential explanation to aberrant DNA methylation in disease.

DNA methylation is an important epigenetic mechanism essential for gene regulation and genome stability[1–3]. In mammals, DNA methylation mainly occurs within the symmetric CpG dinucleotides, established by de novo DNA methyltransferases DNMT3A and DNMT3B during gametogenesis and early embryogenesis[4,5]. Subsequently, the DNA methylation landscape continues to evolve throughout development, through a coordinated action between DNMT3A, DNMT3B and maintenance DNA methyltransferase DNMT1[6–8]. Proper regulation of the

DNA methylation activities of DNA methyltransferases across various regions of the genome is critical for transcriptional programming and other cellular processes[9].

To date, two isoforms of DNMT3A have been identified: DNMT3A1 and DNMT3A2, differing in a DNMT3A1-unique N-terminal ubiquitin-dependent recruitment region (UDR) that recognizes histone H2A lysine-119 mono-ubiquitylation (H2AK119ub1)[10–14]. On the other hand, both isoforms contain a methyltransferase (MTase) domain preceded

[1]Department of Biochemistry, University of California, Riverside, CA, USA. [2]Biochemistry and Molecular Biology Graduate Program, University of California, Riverside, CA, USA. [3]Department of Genetics and Development and Herbert Irving Comprehensive Cancer Center, Columbia University Irving Medical Center, New York, NY, USA. [4]Department of Chemistry, University of California, Riverside, CA, USA. [5]These authors contributed equally: Emily Vig, Jianbin Chen, Kristjan H. Gretarsson. ✉e-mail: jikui.song@ucr.edu

by a Pro-Trp-Trp-Pro (PWWP) domain and an ATRX-DNMT3-DNMT3L (ADD) domain[4,15], which serve as specific readers for di-methylated histone H3 lysine-36 (H3K36me2) and unmethylated H3 lysine-4 (H3K4me0) respectively[11,16-21]. It has been shown that the H3K4me0 interaction relieves the ADD-mediated DNMT3A autoinhibition, thereby stimulating DNMT3A-mediated DNA methylation in the corresponding genomic regions[22,23]. Likewise, recent studies showed that the presence of a H3K36me2 peptide led to enhanced DNA methylation activity of DNMT3A in vitro[24], although the underlying mechanism is unclear.

DNMT3A-mediated DNA methylation is further regulated by DNMT3-like (DNMT3L) protein during gametogenesis and early embryogenesis[25-28]. DNMT3L harbors an N-terminal ADD domain that reads the H3K4me0 mark[29] and a C-terminal methyltransferase-like (ML) domain that associates with the DNMT3A MTase domain to stimulate the DNA methylation activity of the latter[26,30]. Previous studies on the DNMT3A MTase and DNMT3L ML fragment (DNMT3A^MTase-DNMT3L^ML) and the DNMT3A ADD-MTase and DNMT3L ML fragment (DNMT3A^ADD-MTase-DNMT3L^ML) revealed a 3A-3L-3L-3A linear assembly, mediated by a hydrophilic interface between the DNMT3A subunits (a.k.a. RD interface) and a hydrophobic interface (a.k.a. FF interface) between the DNMT3A and DNMT3L subunits[22,31]. Furthermore, the substrate recognition of DNMT3A-DNMT3L complex is mainly mediated by three structural elements: the catalytic loop (residues 707–731), a loop from the target recognition domain (TRD loop: residues 831–847), and a helical element (residues 881–887) at the RD interface[32,33]. In addition, a crystal structure of the DNMT3A^ADD-MTase-DNMT3L^ML complex revealed an autoinhibitory conformation in which the direct interaction between the ADD and MTase domains of DNMT3A leads to occlusion of its catalytic site and H3K4me0-binding site[22], lending an explanation to the coupling between the H3K4me0 binding and the enzymatic activation of DNMT3A[23]. However, how DNMT3A transits from such an autoinhibitory state to an active state remains elusive.

Mutations of DNMT3A have been associated with a wide spectrum of human diseases and developmental disorders[34-38]. For instance, mutation at the site of R882 within the MTase domain represents a hotspot mutation in acute myeloid leukemia (AML) and clonal haematopoiesis[35]. Likewise, mutations within the ADD domain (e.g. E545G) have been associated with AML, Tatton-Brown-Rahman syndrome and myelodysplastic syndrome (MDS)[38-40]. In addition, mutations within the PWWP domain (e.g. W330R and D333N) have been linked to microcephalic dwarfism[34]. Mechanistically, the R882 mutations promote high-order DNMT3A multimerization, causing a dominant-negative effect in its DNA methylation activity[41-44]. The PWWP mutations were shown to lead to gain-of-function DNA methylation in polycomb targets[11,34,36], in part attributed to the loss of the PWWP-H3K36me2 interaction and the competing chromatin-targeting mechanism underpinned by the interaction between DNMT3A1 UDR and histone H2AK119ub1-marked nucleosome[12-14]. Due to the lack of structure-function characterization of a DNMT3A fragment harboring the PWWP, ADD and MTase domains, a comprehensive understanding of how the mutations within the regulatory domains of DNMT3A contribute to disease progression remains lacking.

To elucidate the functional regulation of DNMT3A, we solved the cryo-EM structure of DNMT3A2 bound to full-length DNMT3L, revealing a multilayered autoinhibition. The DNMT3A PWWP domain dynamically interacts with the ADD and MTase domains to strengthen the previously characterized autoinhibition by the ADD domain. Furthermore, the PWWP domain-mediated autoinhibition leads to occlusion of the H3K36me2-binding pocket and the CpG-engaging TRD loop, thereby coupling the H3K36me2 binding with DNMT3A activation. Through molecular dynamics (MD) simulation of DNMT3A-DNMT3L under apo and histone-bound states, we observed that following relief of the PWWP domain-mediated autoinhibition, the ADD-

H3K4me0 interaction disengages the TRD loop from autoinhibitory interaction, releasing the latter for potential DNA contact. On the other hand, the positioning of the ADD domain remains largely intact, suggesting that engagement with both histone and DNA might be required for dislodging the ADD domain toward full activation of DNMT3A. Importantly, our combined structural, biochemical and genomic methylation analysis demonstrates that disrupting the disease-associated PWWP-ADD interface mutations W330R, D333N and E545G leads to impaired DNMT3A autoinhibition and target specificity, which may contribute to the aberrant DNA methylation patterns in disease associated with these mutations. Together, this study uncovers a hierarchical autoinhibitory regulation of DNMT3A, sheds light onto the mechanism of DNMT3A activation and links the PWWP domain-mediated autoinhibition to disease.

## Results

### Cryo-EM structure of the DNMT3A2-DNMT3L complex
To elucidate the functional interplay between the individual domains of DNMT3A and DNMT3L, we focused on a fragment of DNMT3A2 (residues 281–912) in complex with full-length DNMT3L for structure determination (Fig. 1a). The structure of the DNMT3A2-DNMT3L complex was solved at 3.66-Å overall resolution (Fig. 1b–d and Supplementary Figs. 1, 2 and Supplementary Table 1). We were able to trace the PWWP, ADD and MTase domains for one of the DNMT3A subunits (3A-1), the ADD and MTase domains for the other DNMT3A subunit (3A-2), and the ADD and ML domains for both DNMT3L subunits (Fig. 1f). Each of the DNMT3A subunits harbors a S-adenosyl-homocysteine (SAH) molecule, the by-product of cofactor S-adenosyl-methionine (SAM) (Fig. 1c). The DNMT3A2-DNMT3L complex assumes a heterotetrameric assembly resembling that of the DNMT3A^MTase-DNMT3L^ML complex[31] (Fig. 1b, c), with a root-mean-square deviation (RMSD) of 0.93 Å over 845 aligned Cα atoms (Supplementary Fig. 3a).

As expected, the two DNMT3A subunits of the DNMT3A2-DNMT3L complex are highly similar, with an RMSD of 0.70 Å over 386 aligned Cα atoms (Fig. 1d and Supplementary Fig. 2b). Nevertheless, they differ in the conformational state of the PWWP domain: the PWWP domain in the 3A-1 subunit packs against the ADD and MTase domains into a compact fold, whereas the corresponding region lacks defined density in the 3A-2 subunit (Fig. 1d). In line with this difference, the two DNMT3A subunits show a subtle structural variation in the PWWP-interacting site of the ADD domain, in which residues E469-L475 run in parallel with PWWP residues S319-E323 in the 3A-1 subunit but adopts a helical conformation in the PWWP-disordered 3A-2 subunit (Supplementary Fig. 3b). On the other hand, both DNMT3L subunits show an elongated fold where the ADD domain stacks against the ML domain at an interface distant from the DNMT3A-DNMT3L interface (Fig. 1f). In fact, structural alignment of both DNMT3L subunits with the previously reported DNMT3L molecule bound to a histone H3K4me0 peptide (PDB 2PVC) gave RMSD of 0.96 Å and 0.84 Å over 333 and 307 aligned Cα atoms, respectively, indicative of high structural similarity. Notably, the H3K4me0-binding site of the DNMT3L ADD domain in the DNMT3A2-DNMT3L complex is solvent accessible, poised for potential histone binding (Fig. 1f).

It is worth noting that detailed 3D classification of the DNMT3A2-DNMT3L complex also reveals a population of DNMT3A2-DNMT3L particles lacking the density of the PWWP domain in both DNMT3A subunits (Supplementary Fig. 1), highlighting the dynamic nature of the interaction between the PWWP domain and the ADD-MTase domains of DNMT3A.

### The PWWP domain joins the ADD domain to underpin an autoinhibitory conformation of DNMT3A
Next, we aligned the DNMT3A2-DNMT3L complex with the previously reported DNMT3A^MTase-DNMT3L^ML-DNA complex (PDB 5YX2). The DNA-binding site of DNMT3A MTase domain in the DNMT3A2-

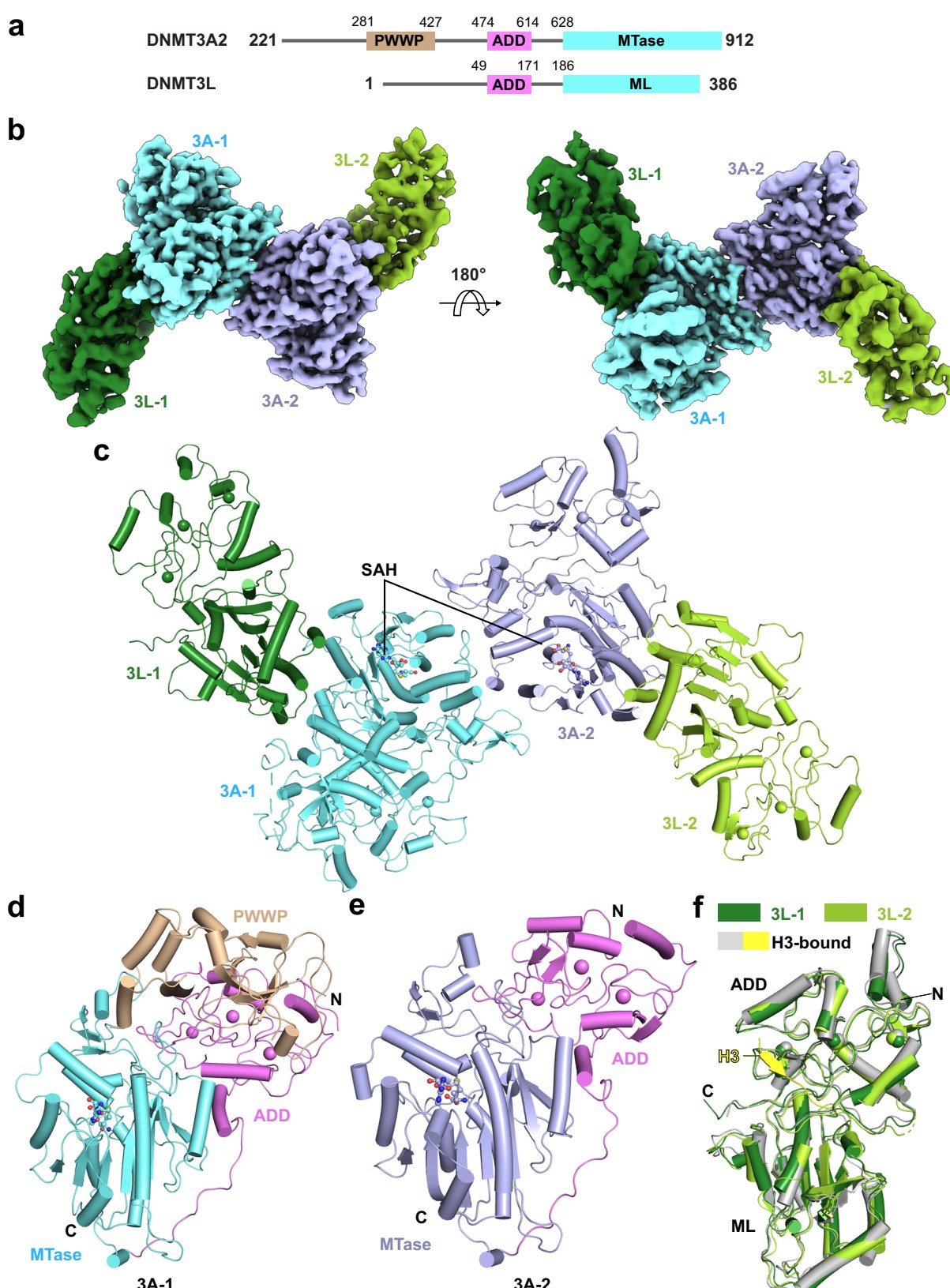

**Fig. 1 | Cryo-EM structure of the DNMT3A2-DNMT3L complex. a** Domain structure of DNMT3A2, with individual domains color coded and delimited by residue numbers. **b** Two opposite views of the density map for the DNMT3A2-DNMT3L complex, with individual subunits color coded. **c** Atomic structure of the DNMT3A2-DNMT3L complex, with individual domains color coded in the same fashion as in (**b**). Unless otherwise indicated, the 3A-1 subunit is used for structural analysis in this study. **d** Atomic structure of the 3A-1 subunit within the DNMT3A2-DNMT3L complex, with individual domains color coded. **e** Atomic structure of the 3A-2 subunit within the DNMT3A2-DNMT3L complex, with individual domains color coded. **f** Structure overlay of the two DNMT3L subunits in the DNMT3A2-DNMT3L complex with the DNMT3L molecule bound to a H3K4me0 peptide (PDB 2PVC). The N- and C-termini of each subunit in d-f are labeled with letter "N" and "C", respectively.

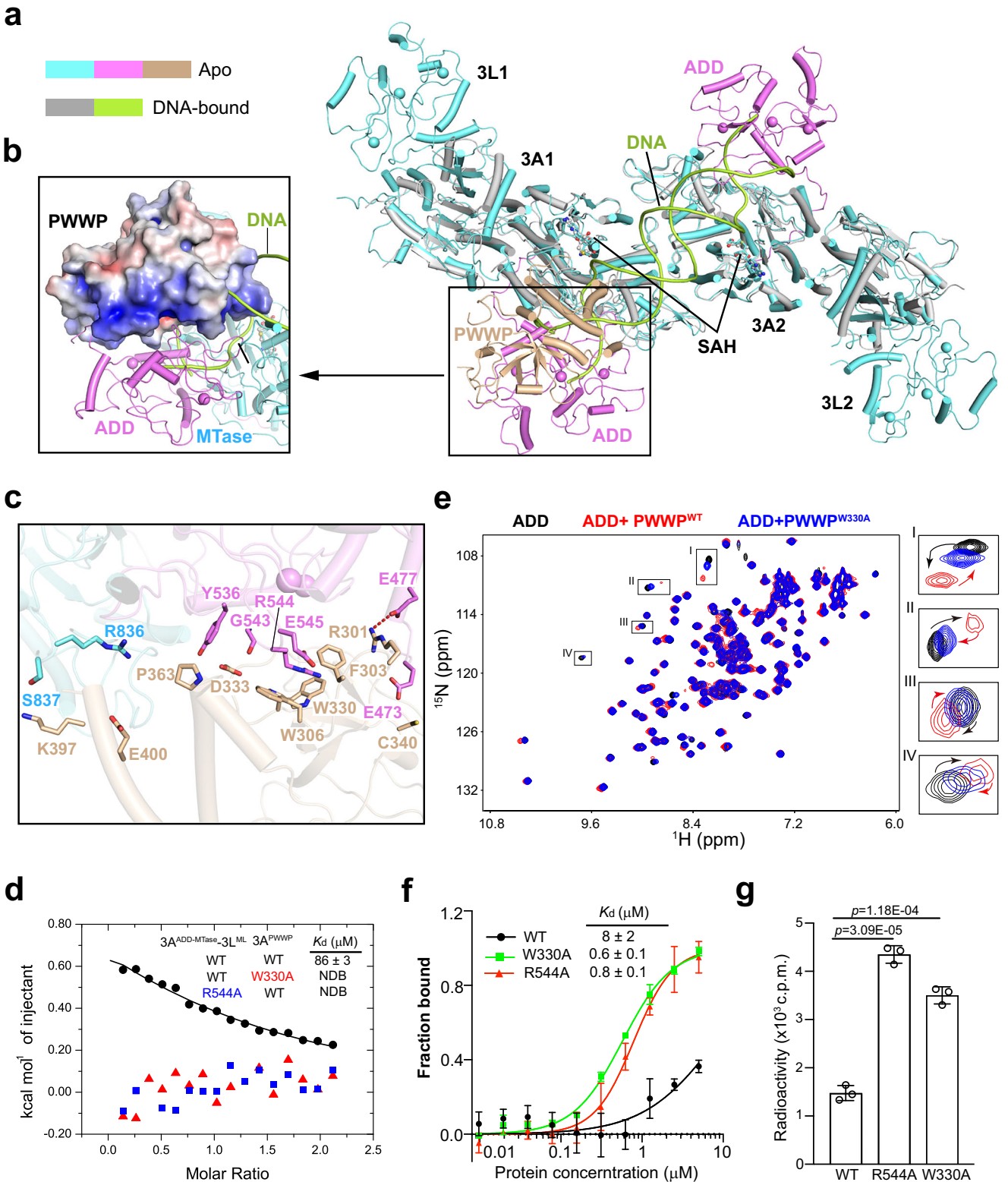

DNMT3L complex is blocked by the ADD domain (Fig. 2a, b), as observed previously[22]. In addition, the DNA binding is blocked by the PWWP domain of the 3A-1 subunit, but not the 3A-2 subunit, indicative of a role for the dynamic PWWP-ADD-MTase interaction in strengthening the autoinhibitory state of DNMT3A (Fig. 2a, b).

Detailed structural analysis of the DNMT3A2-DNMT3L complex reveals that within the 3A-1 subunit, the PWWP domain interacts with the ADD domain as well as the MTase domain (Fig. 2c). At the PWWP-

ADD interface, residue R544 of the ADD domain inserts its sidechain guanidinium group into the aromatic cage of the PWWP domain, lined by residues F303, W306, W330 and D333 (Fig. 2c). Furthermore, residues E473, E477, Y536 and E545 of the ADD domain interact with residues R301, W306, D333, C340, and P363 of the PWWP domain through hydrogen-bonding and/or van der Waals contacts (Fig. 2c). Together, the PWWP-ADD association leads to a buried surface area of ~900 Å². At the PWWP-MTase interface, residues K397 and E400 of the

**Fig. 2 | Structural and biochemical analysis of the DNMT3A PWWP domain-mediated autoinhibition.** Structural overlay between the DNMT3A2-DNMT3L complex and the DNMT3A^MTase-DNMT3L^ML-CpG DNA complex (PDB 5YX2) (**a**), with the steric clash between the PWWP domain (electrostatic surface view) and the DNA molecule highlighted in (**b**). **c** Close-up view of the interaction of the PWWP domain (wheat) with the ADD (violet) and MTase (aquamarine) domains of the 3A-1 subunit, with the interacting residues shown in stick representation. The hydrogen bond is shown as dashed line. **d** ITC binding assays for the DNMT3A^ADD-MTase-DNMT3L^ML complex with the WT or mutant PWWP domain. The $K_d$ and standard deviation were derived from two independent measurements. One representative set of ITC binding data is shown. **e** ^1H-^15N-HSQC spectral overlay of the DNMT3A ADD domain alone and in the presence of WT PWWP domain (PWWP^WT) or W330A-mutated PWWP domain (PWWP^W330A). Four resonance peaks with apparent PWWP^WT-induced chemical shift perturbation are highlighted in the expanded views on the right. The chemical shift perturbations are indicated by the arrows. **f** FP analysis of the interaction between the DNMT3A2-DNMT3L complex, WT or mutant, and a DNA duplex. Data are mean ± s.d. ($n = 3$ biological repeats). **g** In vitro DNA methylation assay of WT or mutant DNMT3A2-DNMT3L complex on 36-mer (GAC)$_{12}$/(GTC)$_{12}$ DNA. Two-tailed Student's $t$ test was used to compare the activity of WT vs mutant. Data are mean ± s.d. ($n = 3$ biological repeats). Source data are provided as a Source Data file.

**a**

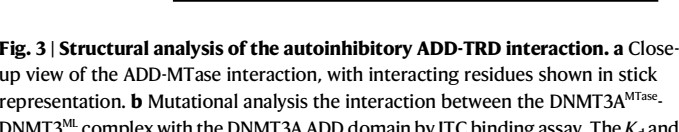

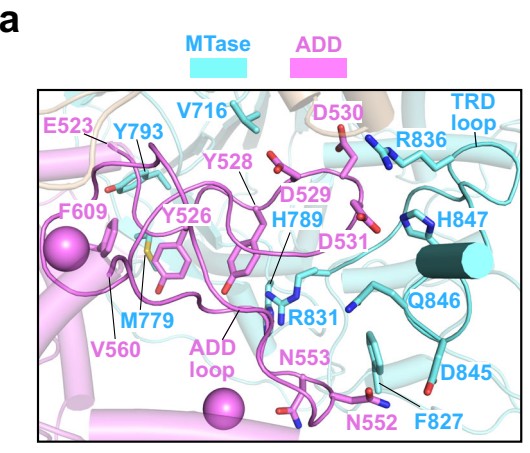

**b**

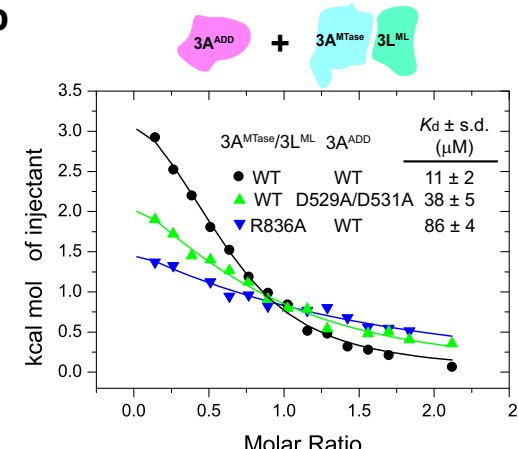

**Fig. 3 | Structural analysis of the autoinhibitory ADD-TRD interaction. a** Close-up view of the ADD-MTase interaction, with interacting residues shown in stick representation. **b** Mutational analysis the interaction between the DNMT3A^MTase-DNMT3^ML complex with the DNMT3A ADD domain by ITC binding assay. The $K_d$ and standard deviation were derived from two independent measurements. One representative set of ITC binding data is shown. Source data are provided as a Source Data file.

PWWP domain are within a distance for van der Waals and/or electrostatic contacts with residues R836-S837 from the TRD loop of the MTase domain (Fig. 2c), a structural element known for CpG-specific DNA recognition[32,33]. As a result of these interactions, the PWWP domain is positioned in front of the substrate-binding site of DNMT3A, blocking the latter from potential DNA binding (Fig. 2a, b).

### Disrupting the PWWP-ADD interaction boosts DNMT3A-mediated DNA methylation

To test the structural observation, we performed isothermal titration calorimetry (ITC) binding assays for the DNMT3A^ADD-MTase-DNMT3L^ML complex and the isolated PWWP domain of DNMT3A. Wild-type (WT) PWWP domain binds to DNMT3A^ADD-MTase-DNMT3L^ML with a dissociation constant ($K_d$) of 86 μM, supporting the structural observation that the PWWP domain directly interacts with the ADD-MTase fragment of DNMT3A. In contrast, no appreciable interaction was observed between the DNMT3A^ADD-MTase-DNMT3L^ML complex and the W330A-mutated PWWP or between the R544A-mutated DNMT3A^ADD-MTase-DNMT3L^ML and WT PWWP (Fig. 2d). We further performed two-dimensional ^1H-^15N Heteronuclear Single-Quantum Correlation (HSQC) NMR experiment for the DNMT3A ADD domain in the absence or presence of the PWWP domain. Spectral comparison of the ADD domain among various conditions reveals that the addition of the PWWP domain leads to notable chemical shift perturbation of several NMR peaks (Fig. 2e), in line with the observed PWWP-ADD interaction. In contrast, replacement of WT PWWP with the W330A-mutated PWWP domain led to a shift of these peaks toward the positions corresponding to the ADD domain alone, indicative of impaired PWWP-ADD interaction (Fig. 2e). These observations lend strong support to the observed PWWP-ADD interaction of DNMT3A.

To examine how the PWWP-ADD interaction affects the DNA binding of DNMT3A2-DNMT3L, we performed the fluorescence polarization (FP) experiment to measure the DNA-binding affinity of WT or mutant DNMT3A2-DNMT3L for a 24-mer DNA duplex. Consistent with a previous observation[22], WT DNMT3A2-DNMT3L binds to the DNA with $K_d$ of 8 μM. In contrast, introducing the R544A or W330A mutation led to >10-fold increase in binding affinity, with $K_d$ of 0.6 μM and 0.8 μM, respectively (Fig. 2f). This observation suggests that disrupting the PWWP-ADD interaction substantially relieves DNMT3A autoinhibition. Consistently, our in vitro DNA methylation assay for WT and mutant DNMT3A2-DNMT3L over 36-mer (GTC)$_{12}$/(GAC)$_{12}$ DNA duplex showed that the R544A and W330A mutations led to a 3.0- and 2.4-fold higher methylation efficiency, respectively. Together, these data support the notion that the intramolecular interaction of the PWWP domain with the ADD and MTase domains gives rise to one layer of autoinhibition for DNMT3A2-DNMT3L.

Sequence analysis of DNMT3A and DNMT3B indicates that the residues involved in the PWWP interaction with the ADD-MTase domains of DNMT3A are largely preserved in DNMT3B (Supplementary Fig. 4), suggesting that the PWWP-mediated autoinhibition likely represents a common regulatory mechanism for the DNMT3A-DNMT3L and DNMT3B-DNMT3L complexes. In support of this notion, a recent study showed that the presence of H3K36me3 peptide leads to enzymatic activation of DNMT3B-DNMT3L[45].

### The ADD domain engages the TRD loop for autoinhibition

Structural analysis of the DNMT3A2-DNMT3L complex reveals that the interaction between the ADD and MTase domains is mainly mediated by a zinc-coordinating loop of the ADD domain (ADD loop: E523-V560) and the TRD loop (Fig. 3a). Notably, the side chain of TRD-loop residue

R836 engages in electrostatic contact with the tri-aspartate motif (D529-D530-D531) of the ADD loop, the side chain of TRD-loop residue R831 engages in cation-π interaction with the aromatic ring of ADD-loop residue Y528, and TRD-loop residues F827, D845, Q846 and H847 are within a distance for van der Waals contact with ADD-loop residues D531, N552 and N553 (Fig. 3a). In addition, the ADD-MTase interaction is supported by van der Waals and hydrophobic contacts involving residues V716, M779, H789 and Y793 from the MTase domain and residues Y526, Y528, D529 and F609 from the ADD domain (Fig. 3a). Collectively, these interactions lead to buried surface area of ~1199 Å² between the MTase and ADD domains. Consistent with the structural observation, our ITC binding assay shows that while the ADD domain binds to the MTase domain with a $K_d$ of 11 μM, consistent with the previous observation[22], introducing the D529A/D531A and R836A mutations reduced the binding by ~3 and ~8 fold, respectively (Fig. 3b).

It is worth noting that the ADD-MTase interface of the DNMT3A2-DNMT3L complex is distinct from what was previously observed for the crystal structure of DNMT3A^ADD-MTase-DNMT3L^ML complex (PDB 4U7P): unlike in the DNMT3A2-DNMT3L complex the ADD-loop approaches the TRD loop for electrostatic contact, in the crystal structure the ADD loop moves away by ~13 Å to engage in electrostatic contact with motif-VIII residues R790 and R792 (Supplementary Fig. 5a–c), reflecting the conformational dynamics of this domain. In addition, analysis of the crystal structure of DNMT3A^ADD-MTase-DNMT3L^ML complex reveals that the ADD domain is involved in crystal packing (Supplementary Fig. 5d). Whether such an effect contributes to the observed structural difference awaits further investigation.

Sequence analysis of DNMT3A and DNMT3B indicates that the residues involved in the ADD-MTase interaction of DNMT3A are largely preserved in DNMT3B (Supplementary Fig. 4). In fact, structural overlay of the DNMT3A2-DNMT3L complex with the previously reported DNMT3B homotetramer (PDB 8EIH)[46] reveals high structural similarity between their central subunits, with an RMSD of 1.46 Å over 406 aligned Cα atoms. Notably, the ADD domains of DNMT3A and DNMT3B are positioned similarly next to the TRD loop (Supplementary Fig. 5e), suggesting that, as with the PWWP interaction, the ADD-MTase interaction is likely evolutionarily conserved between DNMT3A and DNMT3B.

## Functional interplay between the PWWP-H3K36me2 and ADD-H3K4me0 readouts

Structural overlay of the DNMT3A2-DNMT3L complex with the previously reported DNMT3A ADD domain bound to a H3K4me0 peptide (PDB 4QBQ)[19] as well as the DNMT3B PWWP domain bound to a H3K36me3 peptide (PDB 5CIU)[47] mapped the H3K4me0 and H3K36me2-binding sites of the DNMT3A2-DNMT3L complex onto the ADD-MTase interface and the PWWP-ADD interface, respectively (Fig. 4a–c). At the ADD-MTase interface, the H3-binding site is largely accessible for the H3 binding, except that ADD residues D529 and D531 engage in electrostatic contact with TRD-loop residue R836, which competes against their potential salt-bridge interactions with H3K4me0 (Fig. 4b). At the PWWP-ADD interface, the H3K36me2 pocket, formed by PWWP residues F303, W306, W330 and D333, is blocked from potential H3K36me2 binding by ADD residue R544 (Fig. 4c). These observations link the H3K4me0 and H3K36me2/3 binding to the relief of the PWWP-mediated autoinhibition and ADD-mediated autoinhibition of DNMT3A, respectively.

To further elaborate how the PWWP- and ADD-mediated autoinhibition crosstalk with histone bindings, we performed in vitro DNA methylation analysis of DNMT3A2-DNMT3L, WT or mutant, on nucleosome substrates either unmodified or modified with H3 lysine-4 tri-methylation analogue (H3Kc4me3, denoted as H3K4me3 herein) or H3 lysine-36 di-methylation analogue (H3Kc36me2, denoted as H3K36me2 herein)[48]. On the H3K4me3 nucleosome substrates, the D529A/D531A-, R544A- and W330A-mutated DNMT3A2-DNMT3L show

6.5, 3.3 and 3.2- fold higher DNA methylation activity than WT DNMT3A2-DNMT3L (Fig. 4d), supporting the notion that these residues play a role in DNMT3A autoinhibition. In line with previous observations[22,24,49,50], replacement of the H3K4me3 nucleosome with unmodified (H3K4me0) nucleosome led to 7-fold higher methylation efficiency for WT DNMT3A2-DNMT3L; the presence of H3K4me0/K36me2 dual mark further enhanced the DNA methylation efficiency of DNMT3A by 1.7-fold (Fig. 4d). On the other hand, introducing the D529A/D531A mutation at the ADD-MTase interface reduced the H3K4me0-mediated DNMT3A activation to 1.4-fold, while failing to affect the H3K36me2-mediated DNMT3A stimulation (Fig. 4d). Introducing the R544A or W330A mutation at the PWWP-ADD interface reduced the H3K36me2-mediated stimulation to 1.2-fold (for R544A) or even abolished the H3K36me2-mediated stimulation (for W330A); yet, the H3K4me0-mediated stimulation remains strong for both mutants (6.1-fold for R544A and 5.3-fold for W330A) (Fig. 4d). Finally, introducing the D529A/D531A/W330A triple mutation led to reduced (for H3K4me0) or completely lost (for H3K36me2) stimulation by either histone mark (Fig. 4d). Together, these observations support the notion that the respective interplay between DNMT3A PWWP and ADD domains with H3K4me0 and H3K36me2 marks lead to distinct yet additive functional regulation of DNMT3A.

It is worth noting that introducing the R836A mutation, while reducing the ADD-MTase association (Fig. 3b), led to decrease of DNA methylation activity on all three substrates (Supplementary Fig. 6), consistent with the fact that residue R836 plays a key role in CpG interaction[33].

## MD simulation analysis of DNMT3A activation

The TRD loop has previously been shown to mediate the specific recognition of CpG sites by DNMT3A and DNMT3B, with residue R836 of DNMT3A recognizing the CpG guanine (G1) of the substrate (Fig. 5a)[32,33,51]. In this context, structural overlay of the DNMT3A2-DNMT3L complex with the DNMT3A^MTase-DNMT3L^ML-CpG DNA complex (PDB 5YX2) reveals strong steric clashes between the CpG DNA and the ADD and PWWP domains (Fig. 5a), suggesting that the PWWP-ADD-MTase interaction blocks TRD-residue R836 from potential contact with the CpG DNA.

Next, we sought to elucidate the activation mechanism of DNMT3A via MD simulation. Toward this, we generated a structural model of the apo-form DNMT3A^ADD-MTase-DNMT3L (apo-DNMT3A^ADD-MTase-DNMT3L) to mimic the functional state of DNMT3A2-DNMT3L in which the PWWP domain-mediated autoinhibition is relieved. In addition, a structural model of the DNMT3A^ADD-MTase-H3K4me0-DNMT3L complex was generated based on the structures of DNMT3A2-DNMT3L and DNMT3A ADD-H3K4me0 (PDB 4QBQ). The structural models were analyzed after 200-ns MD simulation (Fig. 5b, c). We found that the DNMT3A ADD domain remains positioned similarly between the DNMT3A2-DNMT3L complex, apo-DNMT3A^ADD-MTase-DNMT3L and the DNMT3A^ADD-MTase-H3-DNMT3L complex (Fig. 5b). However, the TRD loop undergoes a notable movement away from the ADD domain, resulting in increased solvent exposure (Fig. 5b). Consistently, our root-mean-square fluctuation (RMSF) analysis of the three structural models reveals that the RMSF of the TRD loop is the highest in the DNMT3A^ADD-MTase-H3-DNMT3L complex but the lowest in the DNMT3A2-DNMT3L complex under the PWWP domain-mediated autoinhibition (Fig. 5c). These observations suggest that relieving the PWWP domain-mediated autoinhibition and gaining the ADD-H3K4me0 interaction would lead to disengagement of TRD residue R836 from the autoinhibitory interaction, allowing the TRD loop to turn away from the ADD domain for potential DNA contact. On the other hand, the observation that the ADD domain remains positioned in place after the H3K4me0 binding implies that the interaction with both histone and DNA is likely required for the full activation of DNMT3A.

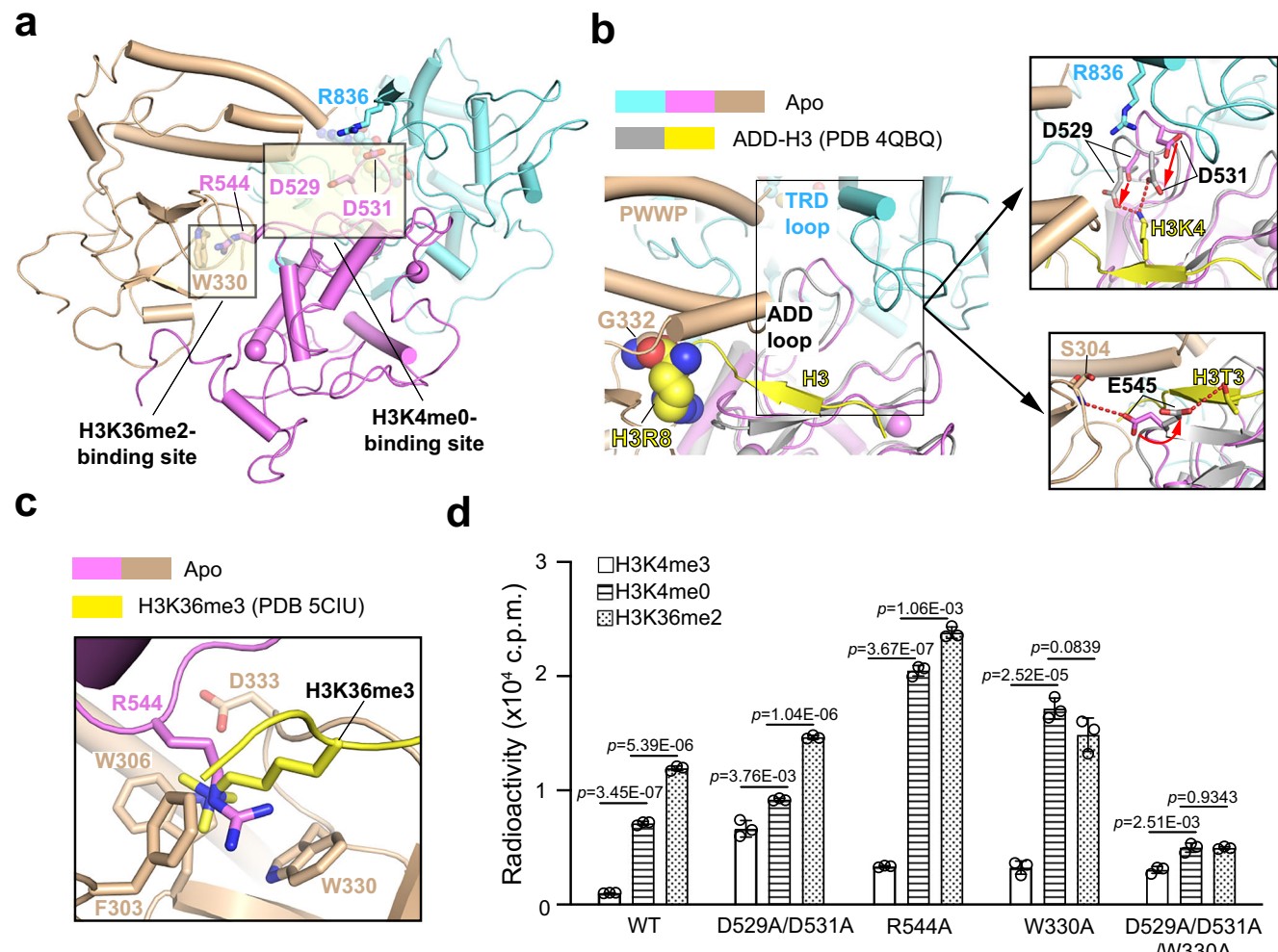

**Fig. 4 | The PWWP- and ADD-mediated intramolecular interactions constitute two distinct layers of autoinhibition. a** Structure of the 3A-1 subunit in the DNMT3A2-DNMT3L complex, highlighting the H3K36me2- and H3K4me0-binding sites located at two separate domain interfaces of DNMT3A. The key residues involved in occluding the histone-binding sites are shown in stick representations. **b** Close-up view of the structural overlay between DNMT3A2-DNMT3L and the H3K4me0-bound DNMT3A ADD (PDB 4QBQ), with the conformational shift of ADD D529, D531 and E545 indicated by red arrow in the expanded views. PWWP G332

and H3 R8 involved in steric clashes are shown in sphere representation. **c** Close-up view of the structural overlay between DNMT3A2-DNMT3L and the H3K36me3-binding DNMT3B PWWP (PDB 5CIU), highlighting the competing interaction between PWWP-H3K36me3 and PWWP-ADD. **d** In vitro DNA methylation assay of DNMT3A2-DNMT3L, WT or mutant, on the nucleosome substrates with various histone modifications. The statistical analysis used two-tailed Student's *t* test. Data are mean ± s.d. ($n = 3$ biological repeats). Source data are provided as a Source Data file.

## Functional analysis of the autoinhibition-disrupting disease mutations

The disease mutations of DNMT3A spread across all the functional domains of DNMT3A[35,37,39,52], including those at the PWWP-ADD interface, such as AML-associated ADD-mutation E545G[39] and microcephalic dwarfism-associated PWWP-mutation W330R and D333N[34] (Fig. 6a). Our $^{1}$H,$^{15}$N-HSQC NMR spectroscopic analysis of the ADD domain in the presence or absence of the PWWP domain reveals that introducing the W330R, D333N and E545G mutations all led to much reduced chemical shift perturbations of the NMR signals of the ADD domain, indicative of impaired PWWP-ADD interaction (Fig. 6b, c). To assess the impact of these mutations on DNMT3A activity, we performed in vitro DNA methylation assays of WT or mutant DNMT3A2-DNMT3L on DNA or nucleosome substrates. Compared with WT DNMT3A2-DNMT3L, the W330R, D333N and E545G mutations all led to increased DNA methylation activity on (GAC)$_{12}$/(GTC)$_{12}$ DNA duplex, reinforcing the notion that these mutations lead to impaired DNMT3A autoinhibition (Fig. 6d). On the other hand, these disease mutants exhibit WT-distinct substrate specificity on the nucleosome substrates: Compared with WT DNMT3A2-DNMT3L, the E545G mutant shows

reduced DNA methylation activity on H3K4me0 and H3K36me2 nucleosomes but increased activity on the H3K4me3 nucleosome substrates, while W330R and D333N mutants both show increased activity on all three nucleosome substrates (Fig. 6e). The caveat of these assays is that they were performed in the presence of excessive amount of substrates, in contrast to the in vivo condition where the enzyme-substrate association also constitutes a crucial reaction step. Nevertheless, these nucleosome context-dependent activity alterations of DNMT3A by the PWWP-ADD interface mutations provide an explanation for the aberrant DNA methylation patterns associated with these mutations in disease.

The DNMT3A W330R mutation has previously been shown to shift the DNMT3A1-mediated DNA methylation to CpG islands (CGIs), in part attributed to abolished PWWP-H3K36me2 binding and the interaction between the DNMT3A1 UDR and the H2AK119ub1-modified nucleosome[10–14]. In light of the observation above that the W330R mutation also leads to disruption of the PWWP-mediated autoinhibition, we asked whether this mutation causes an increase of DNA methylation in the genomic regions beyond the CGIs. To address this, we analyzed the Reduced Representation Bisulfite Sequencing (RRBS)

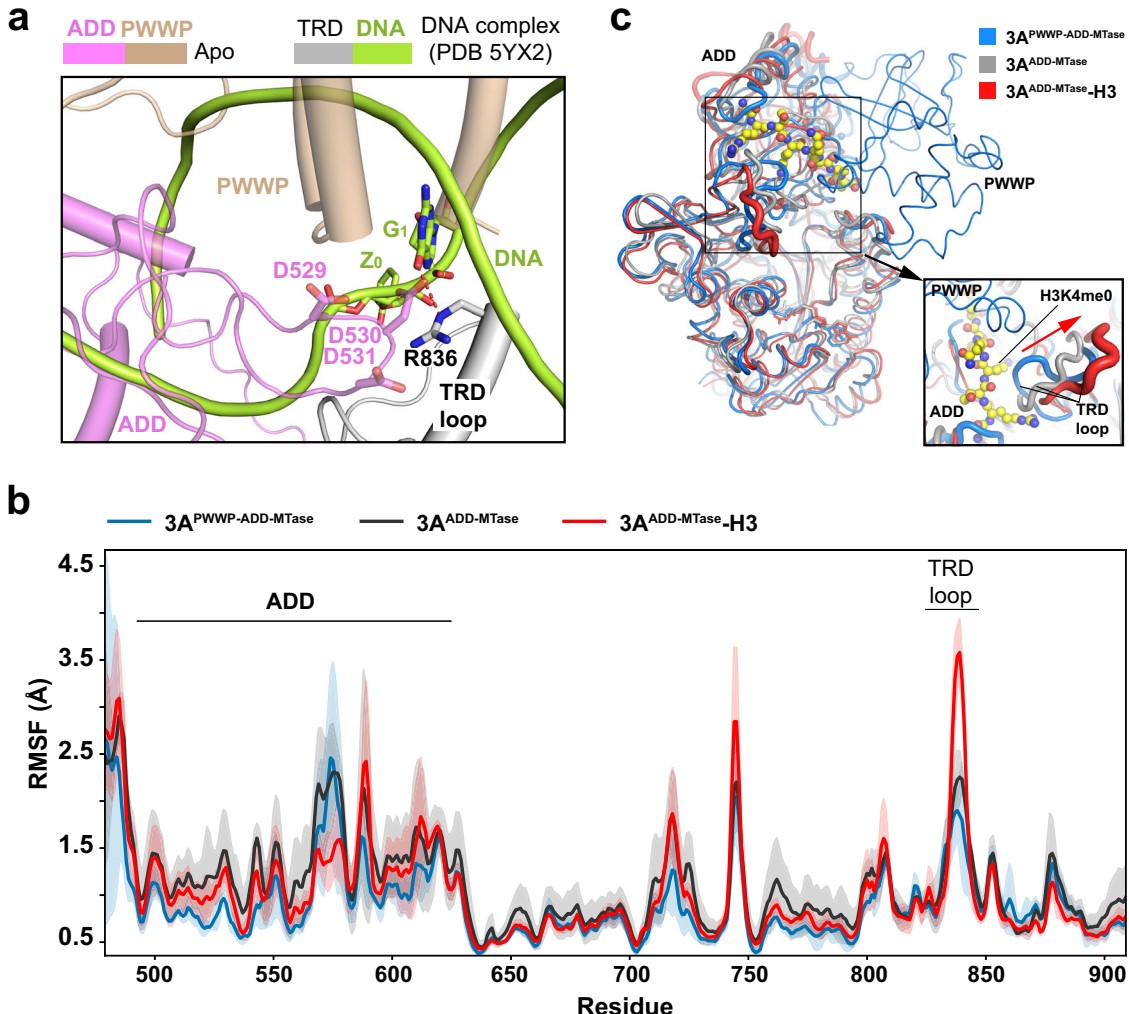

**Fig. 5 | MD simulation analysis of the mechanism of DNMT3A activation.**
**a** Close-up view of the structural overlay between the DNMT3A2-DNMT3L complex and the DNMT3A$^{MTase}$-DNMT3L$^{ML}$-CpG DNA complex (PDB 5YX2), highlighting that the competition between the PWWP-ADD-TRD interaction and the TRD-DNA interaction. The TRD loop residue R836 interacts with ADD residues D529-D531 in the DNMT3A2-DNMT3L complex but CpG guanine (G$_1$) in the DNA-bound form. The cytosine analogue used for structure determination, Zebularine, is labeled as Z$_0$. The hydrogen bond between R836 and G$_1$ is shown as a dashed line. **b** The RMSF values obtained for the ADD-MTase region in the structural model of the intact methylome, focusing on non-CGI regions, in mouse embryonic stem cells (mESCs) with *Dnmt1/Dnmt3A/Dnmt3B* triple knockout (TKO) or *Dnmt1/Dnmt3A/Dnmt3B/Dnmt3L* quadruple knockout (QKO) transduced with WT or W330R DNMT3A1[13]. In line with the previous observation[11,13,34], in the H3K36me2-positive regions, the TKO cells with DNMT3A1 W330R show a lower level of DNA methylation than those with WT DNMT3A1. In contrast, in the H3K36me2-poor regions the W330R cells show a significantly higher level of DNA methylation than WT cells (Fig. 6f). Consistently, in contrast to the activity of WT DNMT3A1 that is positively correlated with the H3K36me2 level, the activity of W330R is largely independent of H3K36me2 and greater than that of WT DNMT3A1 at the low-H3K36me2 regions (Fig. 6g). A similar, although modest, trend of DNA methylation patterns was observed for the QKO cells with WT or W330R DNMT3A1 (Supplementary Fig. 7). These observations suggest that in addition to the impact by the UDR-H2AK119ub1 interaction, disruption of the PWWP domain-mediated DNMT3A autoinhibition by the W330R mutation leads to promiscuous DNA methylation throughout the genome.

DNMT3A2-DNMT3L complex (3A$^{PWWP-ADD-MTase}$), the apo-DNMT3A$^{ADD-MTase}$-DNMT3L (3A$^{ADD-MTase}$), and the DNMT3A$^{ADD-MTase}$-DNMT3L-H3K4me0 complex (3A$^{ADD-MTase}$-H3). Data are mean ± s.d. (*n* = 3 independent runs). The mean RMSF values are shown as solid lines. The RMSF values within 1 s.d. are shown in shade. **c** Sausage view of the RMSF values of the 3A$^{PWWP-ADD-MTase}$, 3A$^{ADD-MTase}$, and 3A$^{ADD-MTase}$-H3, with the conformational transition of the TRD loop indicated by red arrow and the H3K4me0 peptide shown in sphere representation. Source data are provided as a Source Data file.

## Discussion

The functional interplay between the N-terminal domains of DNMT3A and histone marks critically controls the DNA methylation landscape across the genome[11,15]. Here, through structure determination of the DNMT3A2-DNMT3L complex, combined with biochemical, computational and genomic methylation analysis, we deciphered a multi-layered regulation of DNMT3A that links its DNA methylation activity to both H3K4me0 and H3K36me2 marks, with important implications in DNA methylation homeostasis in development and disease.

First, this study discovered that the trilateral PWWP-ADD-MTase interaction leads to occlusion of the H3K36me3-binding site, the H3K4me0-binding site and the CpG-recognition loop of the TRD, thereby providing an autoinhibitory mechanism that couples the H3K4me0 and H3K36me2 bindings respectively with the enzymatic activation of DNMT3A. Conceivably, owing to the competitive nature between the PWWP-ADD and PWWP-H3K36me2 interactions, and between the ADD-TRD, ADD-H3K4me0 and TRD-DNA interactions, both the PWWP and ADD domains would undergo repositioning when

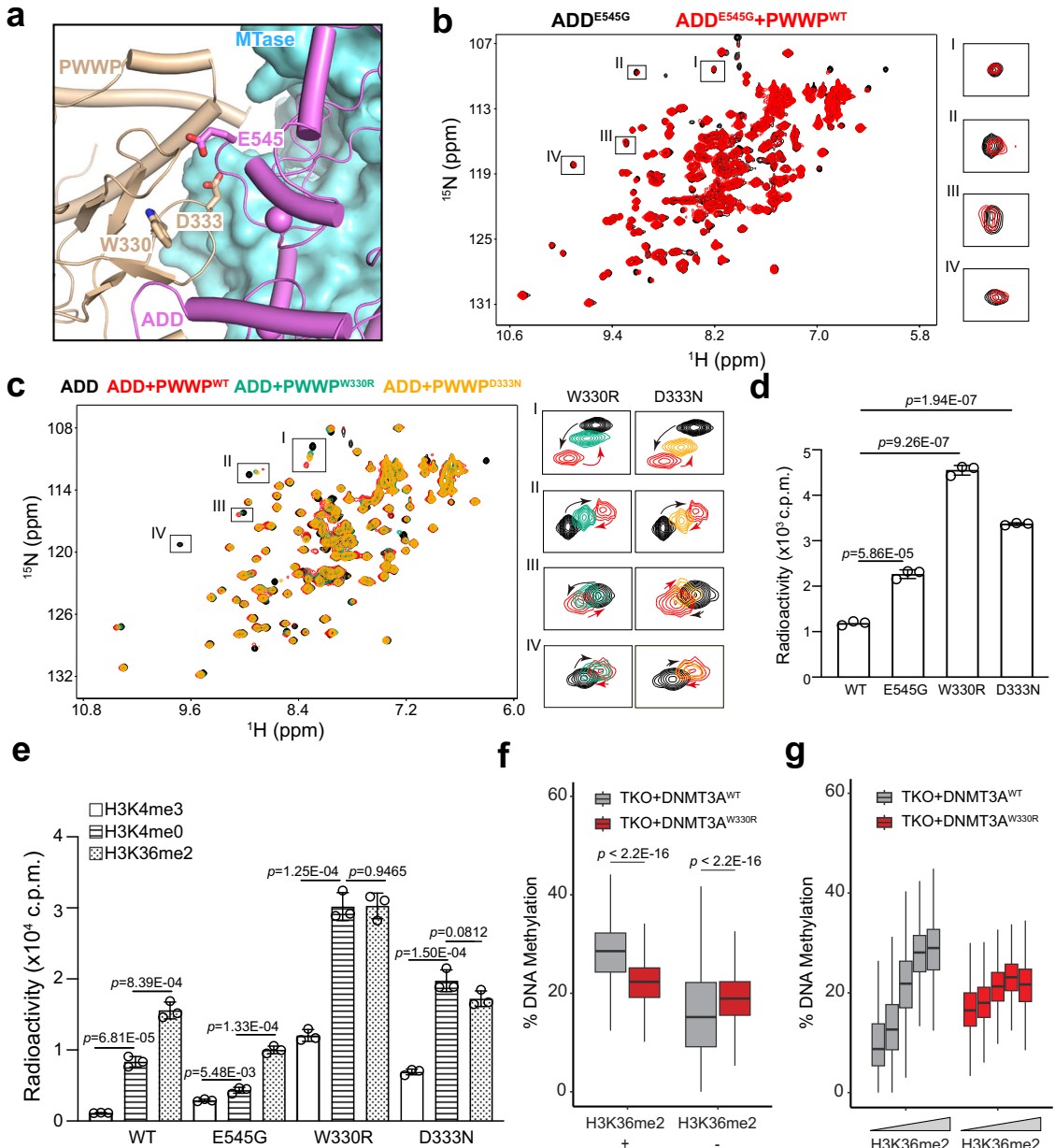

**Fig. 6 | Disease-associated PWWP-ADD interface mutations led to impaired DNMT3A autoinhibition and target selectivity. a** Mapping of the W330R, D333N and E545G to the PWWP-ADD interface, with the PWWP and ADD domains shown in ribbon representation and the MTase domain shown in surface representation. **b** $^{1}H,^{15}N$-HSQC spectral overlay of E545G-mutated DNMT3A ADD, free or in the presence of PWWP$^{WT}$. Four selected regions were shown in expanded views. **c** $^{1}H,^{15}N$-HSQC spectral overlay of the DNMT3A ADD domain, free or in the presence of PWWP$^{WT}$ or mutant DNMT3A PWWP (PWWP$^{W330R}$ and PWWP$^{D333N}$). Four selected regions of the NMR spectra for the W330R (left) or D333N (right) mutants were shown in expanded views. **d** In vitro DNA methylation assay of the DNMT3A2-DNMT3L complex, WT or disease mutant, on 36-mer (GAC)$_{12}$/(GTC)$_{12}$ DNA. Two-tailed Student's t test was used to compare the activity of WT vs mutant. Data are mean ± s.d. (*n* = 3 biological repeats). **e** In vitro DNA methylation assay of

DNMT3A2-DNMT3L, WT or disease mutant, on the nucleosome substrates with various histone modifications. The statistical analysis used two-tailed Student's *t* test. Data are mean ± s.d. (*n* = 3 biological repeats). **f** Box plots showing the DNA methylation levels of 10-kb bins for non-CGI overlapping H3K36me2 positive (left) or negative (right) regions in TKO mESCs reconstituted with WT or W330R DNMT3A1 (*n* = 2 biological replicates). Box-and whisker plots depict 25–75% in the box, whiskers are 1.5 times the interquartile range (IQR), and median is indicated. Wilcoxon rank-sum two-sided test is used for statistical analysis. **g** The DNA methylation level of 10-kb bins that overlapped with quintile Q1–Q5 of non-CGI overlapping H3K36me2 in WT or W330R DNMT3A1 TKO mESCs (*n* = 2 biological replicates). Box-and-whisker plots depict 25–75% in the box, whiskers are 1.5 times the interquartile range (IQR), and median is indicated. Source data are provided as a Source Data file.

transiting into an active state of the DNMT3A-nucleosome complex (Fig. 7). The fact that H3K36me2- and H3K4me0-binding sites are located to two discrete domain interfaces permits a differential impact by various combinations of H3K4me0 and H3K36me2 marks. Furthermore, the structure of the DNMT3A2-DNMT3L complex reveals that the PWWP domain is structurally ordered in one of the DNMT3A subunits but disordered in the other, while the ADD domain assumes a

stable conformation in both DNMT3A subunits. Such a distinct conformational dynamics between the PWWP and ADD domains implies a differential energy barrier for relief of the autoinhibitory regulations by these two domains, providing a mechanism for fine-tuning the enzymatic activity of DNMT3A2-DNMT3L across various genomic regions (Fig. 7). The fact that the residues involved in the PWWP-ADD-MTase interactions are highly conserved in DNMT3B implies that such

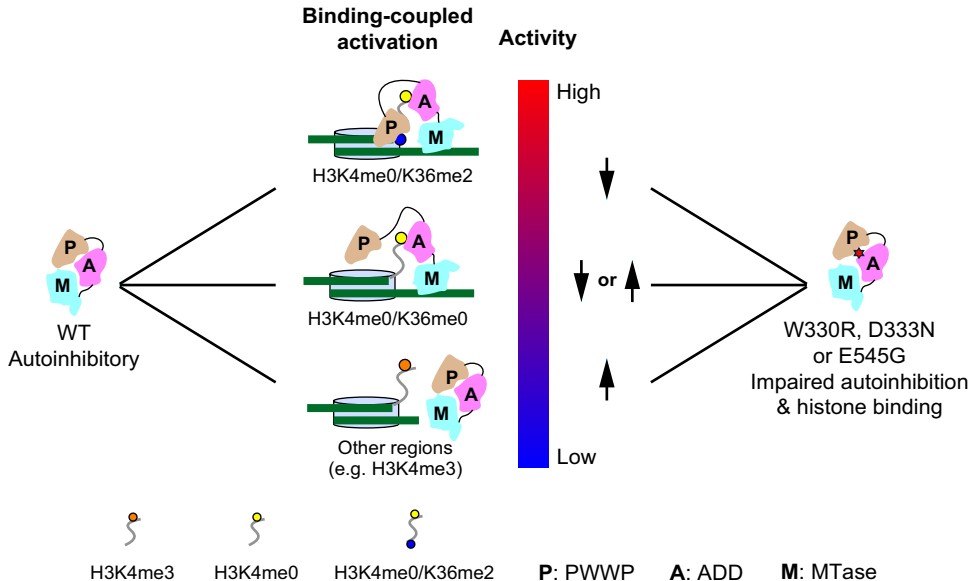

**Fig. 7 | A model for the functional regulation of DNMT3A.** Under the hierarchical autoinhibitory regulation, DNMT3A shows the highest activity on H3K4me0/K36me2-modified nucleosomes but the lowest activity on H3K4me3-modified nucleosomes. The disease-associated PWWP-ADD interface mutations lead to impaired DNMT3A autoinhibition and target selectivity, such as decreased activity on the H3K4me0/K36me2 region but increased activity on the H3K4me3 region, which may contribute to disease progression.

a hierarchical autoinhibition by the PWWP-ADD tandem domains may also apply to the DNMT3B-DNMT3L complex, providing an explanation to the observation that DNMT3B-DNMT3L is subject to enzymatic activation by the bindings of both H3K4me0 and H3K36me3[45].

The PWWP, ADD and MTase domains are commonly present in DNMT3A1 and DNMT3A2, permitting both DNMT3A isoforms to be regulated by the PWWP-H3K36me2 and ADD-H3K4me0 interactions. Previous studies have shown that the H3K36me2 binding-associated disease mutations, such as W330R and D333N, cause a loss of H3K36me2-binding but an increase of DNA-binding affinity of the PWWP domain, which contributes to aberrant DNA methylation activity of DNMT3A in cells[11,13,34]. Furthermore, the interaction between DNMT3A1 UDR and the nucleosome marked with H2AK119ub1 contributes to DNA hypermethylation at polycomb targets, such as CGIs[10–14]. Here, we found that these mutations, along with the E545G mutation from the ADD domain, disrupt the PWWP-ADD interaction, leading to impairment of the PWWP domain-mediated DNMT3A autoinhibition and loss of substrate specificity in vitro, which may contribute to the aberrant DNA methylation patterns in disease. Therefore, this study uncovers a link between the PWWP domain-mediated DNMT3A autoinhibition and pathological consequences of these mutations. Future investigation of such DNMT3A autoinhibition, e.g. via a mutation that disrupts autoinhibition but not any other regulations of DNMT3A, may further clarify this link.

In addition, this study sheds light onto the mechanism of DNMT3A activation. Our MD simulation analysis reveals that the binding of ADD residues D529 and D531 with H3K4me0 releases the TRD loop for potential DNA interaction and eventual DNMT3A activation. Meanwhile, the ADD domain remains in a position that blocks the DNA-binding cleft, implying that full activation of DNMT3A requires both histone and DNA bindings. The requirement of histone and DNA dual binding for enzymatic activation was similarly observed for DNMT1 activation, in which both the interaction between the DNMT1 RFTS domain and histone H3 ubiquitylation at lysine 18 and 23 and the binding to hemimethylated DNA are required to dislodge the RFTS domain from the DNA-binding site of DNMT1[53]. Together, these observations highlight a common regulatory principle for DNA methylation machinery.

## Methods
### Protein expression and purification
The DNA encoding human DNMT3A PWWP-ADD-MTase fragment (residue 281–912) and DNMT3L (residue 1–386) were separately inserted into an in-house bacterial expression vector in which the inserted gene is preceded by a hexa-histidine (His$_6$)-MBP tag and a TEV cleavage site. BL21(DE3) RIL cells harboring the expression plasmids grew at 37 °C and were induced by addition of 0.13 mM (for DNMT3A) or 0.2 mM (for DNMT3L) isopropyl β-D-1-thiogalactopyranoside (IPTG) with 50 μM ZnCl$_2$ when the cell density reached A$_{600}$ of 1.0. The cells continued to grow at 16 °C overnight. The DNMT3A and DNMT3L cells were collected and co-lysed in a buffer containing 50 mM Tris-HCl (pH 8.0), 1 M NaCl, 25 mM Imidazole, 10% glycerol, 10 μg/mL DNase I and 1 mM PMSF. Subsequently, the fusion proteins were purified through a nickel column, followed by removal of His$_6$-MBP tag by TEV cleavage, HiTrap Heparin HP (Cytiva) ion-exchange chromatography and size-exclusion chromatography on a 16/600 Superdex 200 pg column (Cytiva) in a buffer containing 20 mM Tris (pH 8.0), 250 mM NaCl, 5% Glycerol, and 5 mM DTT. The mutants were generated by site-directed mutagenesis and purified in the same way as described above. The primers used in this study were summarized in Supplementary Table 2

The DNA encoding DNMT3A$^{MTase}$ (residue 628–912) and DNMT3A$^{ADD-MTase}$ (residue 469–912), WT or mutants, and DNMT3L$^{ML}$ (residue 178–386) were inserted in tandem into a modified pRSFDuet-1 vector (Novagen), preceded by a His$_6$-Sumo tag and a ULP1 cleavage site. Expression and purification of the DNMT3A$^{MTase}$–DNMT3L$^{ML}$ and DNMT3A$^{ADD-MTase}$–DNMT3L$^{ML}$ complex followed the previously described protocol[33], involving nickel chromatography, ion-exchange chromatography on a HiTrap Heparin HP column and size-exclusion chromatography on a 16/600 Superdex 200 pg column (Cytiva), pre-equilibrated with a buffer containing 20 mM Tris (pH 8.0), 250 mM NaCl, and 5 mM DTT. For analysis of domain interactions, DNMT3A PWWP (residues 281–427), ADD (residues 476–614) and mutants were each cloned into the modified pRSFDuet-1 vector, and expressed and purified following the previously described protocol[46]. In essence, the DNMT3A PWWP domain was purified sequentially through nickel chromatography, ion-exchange chromatography on a HiTrap Heparin HP column and size-exclusion chromatography on a 16/600 Superdex

75 pg column (Cytiva), pre-equilibrated with a buffer containing 20 mM Tris (pH 8.0), 300 mM NaCl and 5 mM DTT. The DNMT3A ADD domain was purified sequentially through nickel chromatography, ion-exchange chromatography on a HiTrap Q HP column (Cytiva) and size-exclusion chromatography on a 16/600 Superdex 75 pg column (Cytiva), pre-equilibrated with a buffer containing 20 mM Tris (pH 8.0), 300 mM NaCl and 5 mM DTT. For the NMR experiments, $^{15}$N-labeled DNMT3A ADD domain was produced in M9 minimal media supplemented with $^{15}$NH$_4$Cl as sole nitrogen resource and purified as described above. All the purified protein samples were concentrated and stored in -80 °C freezer.

### Cryo-EM sample preparation and data acquisition

For cryo-EM sample preparation, concentrated DNMT3A2-DNMT3L complex was subjected to size-exclusion chromatography on a Superdex 200 increase 10/300 column (Cytiva) pre-equilibrated with a buffer containing 20 mM Tris–HCl (pH 8.0), 100 mM NaCl, 5% glycerol and 5 mM DTT. The peak fraction, without treatment by chemical crosslinker, was used for subsequent handling. An aliquot of 2.5 μL of DNMT3A2-DNMT3L sample at a concentration of -1.23 mg/ml was applied to a Quantifoil holey carbon grid (Au, R1.2/1.3, 300 mesh). The grids were glow-discharged for 1 min with H$_2$/O$_2$ through Gatan Plasma clean System (SOLARUS) before use. The grids were blotted and plunge-frozen in liquid ethane cooled by liquid nitrogen with a Vitrobot IV (Thermo Fisher) at 8 °C under 100% humidity. The frozen grids were stored in liquid nitrogen before use. High-resolution cryo-EM data were collected on a Titan Krios electron microscope operating at 300 keV, equipped with a Falcon IV at Pacific Northwest Center for Cryo-EM (PNCC). Movies were recorded at a nominal magnification of ×130,000 with a pixel of 0.7675 Å with the defocus range of −0.8 to −2.3 μm. Each micrograph was recorded with 29 frames with a total dose rate of around 50 $e^-$/Å$^2$.

### Cryo-EM data processing and model building

The cryo-EM data were processed using cryoSPARC (v4.4.1)[54]. The movies were motion-corrected and dose-weighted using the patch-based motion correction module. The contrast transfer function (CTF) of the resulting images was then subjected to patch-based estimation. Automatic particle picking was performed using the TOPAZ method[55]. In total 862,300 particles were extracted from 4756 images, with a down-scaled pixel size of 3.21 Å. After two rounds of clean-up by 2D classifications, those classes with identifiable secondary or tertiary structures, comprised of 527,682 particles, were selected. Initial models were then generated by 3D ab initio reconstruction, followed by one round of heterogeneous refinement ($K=4$) applying C1 symmetry. The class with well-defined density for the PWWP domain was re-extracted with a pixel size of 1.203 Å and subjected to another round of heterogeneous refinement ($K=3$). Finally, 88,258 particles were selected for non-uniform and CTF refinement, resulting in a map with a resolution of 3.64 Å. To improve the density of the PWWP domain, the refined map was used to perform template-based particle picking and 2,634,423 particles were extracted with down-scaled pixel size of 3.21 Å. The extracted particles were evenly divided into six subgroups, while 88,258 before mentioned high-quality particles were treated as "seed" particles and a round of seed-facilitated 3D classification[56] was performed. After removing duplicate particles, 876,165 high-quality particles were selected, re-extracted with a pixel size of 1.203 Å and subjected to two rounds of heterogeneous refinement ($K=3$ and $K=4$). Then 209,307 "good" particles were selected. Subsequently, a 3D focused classification with a mask applied to the density of the ADD domain and PWWP was conducted ($K=4$), and 65,578 particles with identified PWWP features were selected for the final non-uniform and CTF refinement. The final map was sharpened with uniform B factor of −116.28Å$^2$ for further model building and deepEMhancer[57] for visualization. The resolution for the DNMT3A-3L

tetramer was 3.66 Å as given by the Fourier shell correlation criteria (FSC 0.143).

For model building, the reported crystal structures of the DNMT3A$^{ADD-MTase}$ (PDB 4U7P), PWWP (PDB 3LLR) and DNMT3L (PDB 2PV0) were manually fit into the sharpened map in Chimera (v1.17.3)[58]. The initial structural model was then subjected to iterative model building using Coot (v0.9.8)[59] and real-space refinement using Phenix (v 1.21.2_5419)[60]. For the structure, a model-map Fourier shell correlation was calculated using the criterion of 0.5.

### Isothermal Titration Calorimetry (ITC) binding assay

WT DNMT3A$^{MTase}$-DNMT3L$^{ML}$, DNMT3A$^{ADD-MTase}$-DNMT3L$^{ML}$, ADD, PWWP and mutants were dialyzed at 4 °C overnight against a buffer containing 20 mM Tris-HCl (pH 7.5), 100 mM NaCl, and 1 mM β-mercaptoethanol. The final concentrations, determined based on ultraviolet absorption at 280 nm, were 30–50 μM for DNMT3A-DNMT3L tetramer and mutants and 0.3–0.5 mM for the PWWP and ADD domains, WT or mutant.

A MicroCal iTC200 system (GE Healthcare) was used to conduct ITC measurements. A total of 15–17 injections with a spacing of 180 s and a reference power of 5 μcal/s were performed after the temperature was equilibrated to 4 °C. The ITC curves were processed with software Origin (MicroCal) using one-site fitting model.

### Preparation of chemically modified nucleosome

Histones (H2A, H2B, WT or K4C/C110S or K36C/C110S-mutated H3, and H4) from Xenopus laevis were expressed and purified under denatured condition as previously described[12]. Briefly, WT and mutant histones were expressed in *Escherichia coli* BL21 (DE3) RIL at 37 °C. The cells were induced by addition of 0.4 mM IPTG when the cell density reached A$_{600}$ of 1.0 and continued to grow 1 h. The cells were harvested and lysed in a buffer containing 50 mM Tris-HCl (pH 8.0), 0.2 M NaCl and 1% Triton X-100. Histone inclusion bodies were solubilized and denatured in 20 mM Tris-HCl (pH 7.5), 7 M guanidine hydrochloride and 10 mM DTT. Subsequently, denatured histones were purified through sequential ion-exchange chromatography on Q-XL and SP-HP columns (Cytiva) using the buffer containing 20 mM Tris-HCl (pH 7.5), 7 M urea and 2 mM β-mercaptoethanol with a KCl gradient of 0 to 1 M. The purified histone proteins were thoroughly dialyzed against distilled water containing 2 mM β-mercaptoethanol, lyophilized and stored at −80 °C. The K4C and K36C-mutated H3 proteins were chemically alkylated to Kc4me3 (denoted as H3K4me3 herein) and Kc36me2 (denoted as H3K36me2 herein) respectively as previously described[48]. In essence, lyophilized histones (5–10 mg) were each dissolved in -1 mL of alkylation buffer containing 1 M Hepes (pH 7.8), 4 M guanidine hydrochloride, 10 mM D/L-methionine, and then mixed with 20 μL of 1 M DTT and incubated at 37 °C for 1 h. Alkylation of H3K$_C$36 was carried at room temperature, with the reaction mixture subject to sequential additional of 50 μL of 1 M (2-chloroethyl)-dimethylammonium chloride (Sigma-Aldrich), 10 μL of 1 M DTT, and 50 μL of 1 M (2-chloroethyl)-dimethylammonium chloride. Alkylation of H3K$_C$4 was performed at 50 °C, with the reaction mixture subject to sequential addition of 100 mg (2-bromoethyl) trimethylammonium bromide and 10 μL of 1 M DTT. All alkylation reactions were quenched by addition of β-mercaptoethanol (50 μL, 14.2 M), followed by dialysis against distilled water containing 2 mM β-mercaptoethanol. The histone octamer was reconstituted by mixing four histones with equal molar and dialysis in refolding buffer (20 mM Tris-HCl pH 7.5, 2 M NaCl, 1 mM EDTA and 5 mM 2-mercaptoethanol) and further purified by size-exclusion chromatography on a HiLoad 16/600 Superdex 200 pg column.

The Widom 601 DNA flanked by a 35-bp DNA sequence (5'-GTCGTCGTCGTCGTCGTCGTCGTCGTCGTCGTCGT<u>ATCGAGAATCCCGGTGCCGAGGCCGCTCAATTGGTCGTAGACAGCTCTAGCACCGCTTAAACGCACGTACGCGCTGTCCCCCGCGTTTTAACCGCCAAGGGGATTACTCCCTAGTCTCCAGGCACGTGTCAGATATATACATCCGAT</u>-3'; Widom 601

DNA sequence is underlined) was PCR amplified and purified as previously described[12]. To assemble the nucleosome, purified histone octamer was mixed with the 601 DNA in a 1:1 molar ratio, followed by stepwise dialysis against a buffer containing 10 mM Tris-HCl (pH 8.0), 1 mM EDTA, 1 mM DTT and 0.25–2 M KCl. Finally, the assembled NCP was dialyzed against a buffer containing 10 mM Tris-HCl (pH 8.0) and 1 mM DTT overnight. The assembled NCP was analyzed in 5% Tris-borate-EDTA (TBE) native gel in 0.2× TBE buffer (89 mM TBE, pH 8.3). The DNA band was visualized using SYBR Gold staining and scanned using a ChemiDoc imager (Bio-Rad).

## In vitro enzymatic assay

For comparison of the DNA methylation activity between WT and mutant DNMT3A-DNMT3L on nucleosomes, a 20-µL reaction mixture contained 0.1 µM DNMT3A-DNMT3L, 0.4 µM assembled nucleosome, 0.55 µM S-adenosyl-L-[methyl-$^3$H] methionine (specific activity 87.9 Ci/mmol, PerkinElmer), 0.65 µM non-radioactive AdoMet in 50 mM Tris–HCl (pH 8.0), 75 mM NaCl, 0.05% β-mercaptoethanol, 0.02% NP-40, 10% glycerol and 200 µg/mL BSA was incubated at 37 °C for 30 min and quenched by addition of 5 µL of 10 mM non-radioactive AdoMet. For detection, 8 µL of the reaction mixture was spotted on Amersham Hybond-N$^+$ paper (Cytiva) and air dried. The paper was then washed with 0.2 M cold ammonium bicarbonate (pH 8.2) (twice), Milli Q water and ethanol. Subsequently, the paper was air dried and transferred to scintillation vials filled with 3 mL ScintiVerse cocktail (Thermo fisher). The radioactivity of tritium was measured with a Beckman LS6500 counter. For the comparison of activity on naked DNA, a 36-mer $(GAC)_{12}/(GTC)_{12}$ DNA duplex was used as the substrate and the reaction condition is similar as above, except that the concentration of DNMT3A-DNMT3L and dsDNA were kept at 0.2 µM and 0.75 µM, respectively. Each methylation assay was carried out in triplicate and plotted in the GraphPad Prism (v.10).

## Fluorescence polarization (FP) assay

For FP assay, 15 nM FAM-labeled 24-mer DNA duplex (upper strand: /5FAM-T/CGGACAGGATGTATATATCTGAC; lower strand: GTCAGA-TATATACATCCTGTCCGA) was mixed with various concentrations (0.01, 0.02, 0.04, 0.08, 0.15, 0.31, 0.62, 1.25, 2.5, and 5 µM) of DNMT3A2-DNMT3L for 30 min at 4 °C in a buffer containing 20 mM Tris-HCl (pH 7.5), 50 mM NaCl, 5% glycerol, and 5 mM DTT. The FP measurements were performed on a ClarioStar plus plate reader (BMG LABTECH) at 25 °C. The DNA bound fractions were calculated as (mP − baseline mP)/(maximum mP − baseline mP), where mP represents the FP value. Each reaction was carried out in triplicate and the bound fractions were fit against protein concentrations using the specific binding model with Hill slope in GraphPad Prism (v.10).

## NMR spectroscopy

Uniformly $^{15}$N-labeled WT or mutant DNMT3A ADD, ADD$^{E545G}$ and PWWP domains were dialyzed overnight against buffer containing 10 mM Tris–HCl (pH 7.2), 100 mM NaCl, and 1 mM DTT and quantified based on ultraviolet absorption at 280 nm. The NMR spectroscopy was performed at 20 °C on a 700 MHz spectrometer (Bruker Avance) equipped with a cryogenic probe in a buffer containing 10 mM Tris–HCl (pH 7.2), 50 mM NaCl, 1 mM dithiothreitol (DTT), and 10% (v/v) D$_2$O. $^1$H,$^{15}$N–HSQC NMR spectra were acquired for 0.2 mM $^{15}$N-labeled DNMT3A ADD in absence or presence of equimolar amounts of PWWP, WT or mutant. Similarly, the $^1$H,$^{15}$N–HSQC NMR spectra of 0.2 mM $^{15}$N-labeled DNMT3A ADD$^{E545G}$ were collected in the absence or presence of equimolar WT DNMT3A PWWP. The NMR spectra were processed using TopSpin (Bruker BioSpin) and analyzed with Mnova NMR (Mestrelab Research).

## Molecular dynamics (MD) simulation

The structural models of the DNMT3A2-DNMT3L, DNMT3A$^{ADD-MTase}$-DNMT3L and DNMT3A$^{ADD-MTase}$-H3-DNMT3L complexes were used for molecular dynamic (MD) simulation of the DNMT3A$^{PWWP-ADD-MTase}$, DNMT3A$^{ADD-MTase}$ and DNMT3A$^{ADD-MTase}$-H3 states. In essence, the structural model of DNMT3A$^{ADD-MTase}$-DNMT3L was generated by removal of the PWWP domain in the DNMT3A2-DNMT3L complex. The structural model of DNMT3A$^{ADD-MTase}$-H3-DNMT3L was obtained based on the structural overlay between the DNMT3A2-DNMT3L complex and the DNMT3A ADD-H3K4me0 complex (PDB 4QBQ). MD simulations were performed using the AMBER20 package[61]. The partial charges of S-adenosylhomocysteine (SAH) were calculated using AM1-BCC. MD simulations were initiated using the ff14SB force field for protein and GAFF2 for ligand. ZAFF forcefield was used to stabilize the zinc finger[62]. Covalent bonds were added to the 4 cysteine nearest to each zinc, for a total of 48 zinc-cysteine covalent bonds. All systems were solved in water boxes 12 Å from the protein edge using TIP3P explicit solvent model at temperature of 298 K with NPT ensemble. Na$^+$ ions were added to neutralize the system. A 12-Å cutoff was used for short-range non-bonded interactions and the long-range electrostatic interactions were computed by the particle mesh Ewald method (PME). The water molecules were minimized for 10,000 steps, followed by minimization of the entire system for 20,000 steps (Supplementary Table 3). The solvated system was equilibrated under constant pressure and temperature (NPT ensemble) at 50 K for 200 ps, from 75 K to 275 K with 25 K increments and 100 ps each, and finally at 298 K for 400 ps. Production runs were also performed in the NPT ensemble at 298 K using a Langevin Thermostat with 2-fs time-steps. We performed 200-ns MD saving every 1-ps interval, then resaved the trajectories every 1 ns, resulting in 200 frames for analysis. To understand the differences in dynamics, simulations were visualized using Visual Molecular Dynamics (VMD) software[63]. The RMSF of each alpha carbon was calculated based on alignment of DNMT3A residues directly in VMD. The RMSF values were averaged with standard deviation across three replicas and plotted versus residue position. The coordinates resulting from the MD simulations were summarized in Supplementary Data 1.

## CUT&RUN and reduced representation bisulfite sequencing (EM-seq) data analysis

H3K36me2 CUT&RUN sequencing data from QKO (*Dnmt1/Dnmt3A/Dnmt3B/Dnmt3L* KO) mESC (GSE247019) was analyzed using the *multiBigwigSummary* function from DeepTools (v3.5.1)[64] to quantify H3K36me2 CUT&RUN signal in 10-kb genome-wide bins. H3K36me2 domains were called using SEACR (v1.3)[65]. Bismark coverage files of TKO (*Dnmt1/Dnmt3A/Dnmt3B* KO) and QKO mESC EM-seq data (DNA Methylation sequencing) (GSE247019) were analyzed using methylkit (v1.26)[66]. All CpGs with at least 2 reads were retained, and the DNA methylation status of 10-kb bins with at least 5 CpGs were retained for analysis. To focus on the non-CGI genomic regions, 10-kb bins overlapping CGIs were left out of the downstream analysis.

## Reporting summary

Further information on research design is available in the Nature Portfolio Reporting Summary linked to this article.

# Data availability

The atomic model for the DNMT3A2-DNMT3L complex has been deposited in the Protein Data Bank under accession code 9PRW. The cryo-EM density map has been deposited in EMDB under the accession number of EMD-71814. The PDB accession codes 2PVC 2PV0 3LLR, 4QBQ, 4U7P, 5CIU, 5YX2 and 8EIH were used in this study. The Cut&Tag and DNA methylation profiling data deposited in NCBI Gene Expression Omnibus under accession number GSE247019[13] were used in this study. Source Data are provided in the Source Data file. Source data are provided with this paper.

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

## Acknowledgements

This work was supported by NIH grants (R35GM119721 to J.S., R35GM138181 and R01CA266978 to C.L., and R01GM109045 to C.A.C.), NSF grant MCB-2437134 to C.A.C. and the Academic Senate Award of UC Riverside to J.S. E.V. and N.K. were supported by GAANN fellowship of Department of Education (P200A210136). N.K. was also supported by the administrative supplement of NIH R35GM119721. A portion of this research was supported by NIH grant U24GM129547 and performed at the PNCC at OHSU and accessed through EMSL (grid.436923.9), a DOE Office of Science User Facility sponsored by the Office of Biological and Environmental Research. Collection of the Cryo-EM data was assisted by Dr. Rose Marie Haynes. We thank Dr. Lingchao Zhu in ACIF NMR facility of UC Riverside for assistance in NMR data collection and UC Riverside High Performance Computer Cluster for providing computation resources.

## Author contributions

J.L., E.V., J.C., K.H.G., N.K. and Z.S. performed the experiments, C.L., C.C. and J.S. supervised the study. J.L. and J.S. wrote the manuscript and all authors approved the manuscript.

## Competing interests

The authors declare no competing interests.
