## [Transparent Peer Review file · Nature Communications]

Structural insight into hierarchical DNMT3A autoinhibition and its dysregulation in disease

Corresponding Author: Dr Jikui Song

Version 0:

Reviewer comments:

Reviewer #1

(Remarks to the Author)

This manuscript reports the cryoEM structure of the hetero-tetrameric DNMT3A-DNMT3L complex. In this structure, one of the DNMT3A subunits displays a well resolved PWWP domain assembled together with the ADD and MTase domains. This arrangement reveals that not only is the DNA-binding loop in the MTase domain blocked by ADD and PWWP domains, but the H3-binding pockets within both the ADD and PWWP domains are also occluded. Based on these observations, the authors propose a hierarchical autoinhibition mechanism that may underlie the differential DNMT3A activity observed on variously modified nucleosomes. Overall, this study provides the first structural visualization of the PWWP, ADD and MTase domains of DNMT3A assembled in an auto-inhibitory conformation. These findings offer mechanistic insight into how DNMT3A adopts its inactive state and explains why disease-linked mutations at the PWWP-ADD interface lead to aberrant DNA methylation. A few specific concerns and suggestions are outlined below.

1. Several previous studies have shown that the PWWP domain is highly flexible in both DNMT3A and DNMT3B. However, this manuscript unexpectedly resolves the structure of the PWWP domain assembled with the ADD and MTase domains in an auto-inhibitory conformation. Although this finding is surprising, the ITC experiment presented in Figure 2d indeed supports interactions between the PWWP domain and the PWWP-truncated DNMT3A-DNMT3L complex. Could the authors elaborate on the possible reasons why the PWWP domain appears less flexible in their study?

2. In the active state, the ADD domain in DNMT3A undergoes a positional shift relative to the MTase domain as demonstrated in previous structural studies. Similarly, the PWWP domain is also expected to change its position in the active state, as suggested by the model shown in Figure 7. To further support this model, do the authors have any evidence indicating that the PWWP domain interacts differently with the PWWP-truncated DNMT3A-DNMT3L complex in the presence of the nucleosome substrates?

Reviewer #2

(Remarks to the Author)

This is an exciting manuscript centered on DNMT3A/L structures that reveal a new autoregulatory mechanism, whose phenomenology has been described, but whose molecular details remained mysterious. The key breakthrough here is the new structural observation of a PWWP domain of DNMT3A packing against the enzyme active site and occluding histone recognition pockets, that is convincingly argued to be a novel autoinhibited state of the complex. The experiments are highly rigorous and combine the strengths of NMR and cryoEM, as well as convincing accompanying biochemistry to define the novel mode of PWWP interaction with the ADD and methyltransferase domains that serves to sharpen the previously noted nucleosome substrate specificity, that was thought to be purely driven by favorable binding enthalpy as opposed to a dynamic interplay of the PWWP domain occluding the active site and histone binding pockets.

This goes beyond merely defining the structural and biochemical basis of H3K36me regulation of DNMT3a/L, the reanalysis

of prior W330R mutant methylation patterns is consistent with the proposed model, but also can be interpreted as reduced affinity for H3K36me3 by ablation of the aromatic cage that binds H3K36me3, a mutant that separates this function from PWWP autoinhibition would be needed to make a more robust conclusion from these data, but given the strength of the in vitro data, I am convinced that their newly discovered mechanism is operant in cells. Several noted disease associated mutations can be reinterpreted in light of the toggling in between the autoinhibited state and the open state, and these are tested in vitro as well.

As this is an extremely important enzyme in cancer, development and epigenetic inheritance, this new regulatory mechanism will be of interest to a broad readership of Nature Communications, and likely extends to the related DNMT3B/L complex.

While the gross features are similar to prior structures, one of the two 3A protomers in the tetramer has the PWWP domain modellable, and it is oriented to block the DNA methyltransferase active site, the ADD domain that binds unmethylated H3 tail, and the PWWP's binding pocket that has been previously shown to bind H3K36me2.

Evidence that the newly discovered autoinhibition interface is real and important is quite strong—mutations, ITC and NMR all support the interpretation that the PWWP-ADD autoinhibited interface observed in the new structure represents state in cells that is likely populated. Moreover, the inhibition of histone binding and methyltransferase activity being relieved by mutants that disrupt the PWWP-ADD and PWWP-MTase interfaces strengthens this interpretation. It is speculated based on TRD loop position and superposition with CpG DNA bound forms of the complex this PWWP packing inhibition mode would also occlude this interface, but it is not explicitly tested beyond and MD simulation (and would be challenging to actually test). Importantly, the authors also show that this autoinhibition mode also sharpens the nucleosomal substrate preferences previously noted in the literature for H3K4me0 and H3K36me2 for the ADD domain and PWWP-domain, respectively. Major point—a mutant that disrupts the autoinhibited state that has clean separation of function from other individual module functions, tested in living cells would improve the case that this mechanism is relevant in cells, but I do not think is essential as the accumulated in vitro data is quite compelling on its own and consistent with prior cell and organism based studies. Apart from a few typos, the paper is well written and figures well made to convey the key structural details, the rigor of the data is high and key points are made by multiple lines of unambiguous evidence.

Minor points:

Methyllysine analogs are used in this work rather to mimic the normal counterparts. Given noted deviation of MLAs from the native modifications observed in other contexts, I think this should be more prominently noted in the main text rather than appearing solely in the methods to avoid confusion.

The Kd reported for the ADD-PWWP interface needs to be appropriately truncated for significant digits (8+/-2 uM). Similar considerations apply to Fig 3b Kds.

A few typos to fix: line 56: "H3K36m2" , line 254 "bindings"

Reviewer #3

(Remarks to the Author)

In this manuscript the authors present a thorough and relevant study that nicely integrates structural information with biophysical, biochemical and molecular modeling studies to offer mechanistic insight into orchestration of the DNMT3A2/DNMT3L association with histone mark recognition, DNA substrate binding, and enzymatic methylation activity. This not only provides key mechanistic insight into the regulation of DNMT3A activity in normal cell function but importantly offers mechanistic perspective for how site-specific disease associated protein mutations disrupt key auto-inhibiting interactions to support observed hypermethylated DNA disease phenotypes. The inclusion of mutational studies in orthogonal biophysical and biochemical assays provides compelling support for the structural studies and the core conclusions presented. Overall, this is a comprehensive, timely and important investigation that advances our understanding of the molecular mechanisms by which epigenetic alterations and aberrant DNA methylation accumulation may occur to promote disease conditions. The findings are highly valuable for advancing the field. Nevertheless, there are a few points that should be addressed prior to publication that would enhance clarity of presentation, strengthening the major findings.

1) In the Abstract, most of the acronyms are defined, but not PWWP and ADD. It is recommended to define these to be consistent with defining all the other acronyms.

2) Fig. 2: In Fig. 2d, as well as 3b, there does not seem to be error bars included on the presented ITC data, which should be added in.

Regarding the presented 1H/15N HSQC in Fig. 2e, it is difficult to differentiate the magenta from red peaks, making it difficult to readily evaluate the author's conclusions. It is also stated in the text that addition of WT PWWP results in "...substantial chemical shift perturbation of several..." ADD NMR resonances, though there seems to be very few ADD resonances that experience a chemical shift perturbation and the majority of the ones that do have small chemical shift perturbations. In addition, it is claimed that addition of the PWWP W330A variant shifts the ADD resonances "...back to.." the non-interacting form. The presented spectra support the conclusion that the ADD resonances move back toward the unbound ADD form, but do not revert back fully. It is thus recommended to attenuate the claims in these two statements. Having said that, it is again difficult to globally assess these claims given the lack of contrast between the red and magenta spectra.

3) The Fig. 4 legend denotes f, rather than d. It is stated in regard to the DNA methylation assay data that neither variant R544A or W330A affected "...H3K4me0-mediated stimulation appreciably...". It is presumed this statement is being made relative to WT, in which it appears that the difference between H3K4me3 and H3K4me0 substrates in activating DNMT3A is greater for both variants than the 7-fold difference observed for the WT.

Version 1:

Reviewer comments:

Reviewer #1

(Remarks to the Author)

This is a well-written manuscript describing the cryoEM structure of the hetero-tetrameric DNMT3A-DNMT3L complex. The authors have satisfactorily addressed the questions I previously raised.

Reviewer #2

(Remarks to the Author)

The revisions have addressed all of my concerns, which were very minor to begin with. I fully endorse its publication in its present form.

Reviewer #3

(Remarks to the Author)

The authors were highly responsive to the Reviewer's concerns and have sufficiently addressed the points raised, leading to an overall improved manuscript. This reviewer has no further concerns that would need to be addressed prior to publication.

General Response

We thank all reviewers for their collective efforts in reviewing our manuscript, their positive view of our work, and their constructive comments for improving the manuscript. As outlined below in the point-by-point response (marked in blue), we have now systematically addressed all the raised critiques and have incorporated them in the revised manuscript (marked in red).

Point-by-point Response

Response to Reviewer 1

This manuscript reports the cryoEM structure of the hetero-tetrameric DNMT3A-DNMT3L complex. In this structure, one of the DNMT3A subunits displays a well resolved PWWP domain assembled together with the ADD and MTase domains. This arrangement reveals that not only is the DNA-binding loop in the MTase domain blocked by ADD and PWWP domains, but the H3-binding pockets within both the ADD and PWWP domains are also occluded. Based on these observations, the authors propose a hierarchical autoinhibition mechanism that may underlie the differential DNMT3A activity observed on variously modified nucleosomes. Overall, this study provides the first structural visualization of the PWWP, ADD and MTase domains of DNMT3A assembled in an auto-inhibitory conformation. These findings offer mechanistic insight into how DNMT3A adopts its inactive state and explains why disease-linked mutations at the PWWP-ADD interface lead to aberrant DNA methylation. A few specific concerns and suggestions are outlined below.

We thank the Reviewer for his/her positive assessment of this work and have addressed his/her concerns below.

1. Several previous studies have shown that the PWWP domain is highly flexible in both DNMT3A and DNMT3B. However, this manuscript unexpectedly resolves the structure of the PWWP domain assembled with the ADD and MTase domains in an auto-inhibitory conformation. Although this finding is surprising, the ITC experiment presented in Figure 2d indeed supports interactions between the PWWP domain and the PWWP-truncated DNMT3A-DNMT3L complex. Could the authors elaborate on the possible reasons why the PWWP domain appears less flexible in their study?

The Reviewer's point is well taken. Indeed, compared with the stable ADD-MTase interaction, the interaction of the PWWP domain with the ADD and MTase domains is rather dynamic: within the DNMT3A homodimer, the PWWP domain is only traceable for one DNMT3A subunit, but not the other; in contrast, the ADD domains in both DNMT3A subunits are well defined. To define the conformational state of the PWWP domain, we

resorted to a “seed”-facilitated 3D classification approach, combined with focused classification with a mask applied to the density of the ADD domain and PWWP. These data processing approaches permitted us to obtain the defined density for the PWWP domain. We have described our data processing pipeline in detail in the Method section and supplementary materials.

2. In the active state, the ADD domain in DNMT3A undergoes a positional shift relative to the MTase domain as demonstrated in previous structural studies. Similarly, the PWWP domain is also expected to change its position in the active state, as suggested by the model shown in Figure 7. To further support this model, do the authors have any evidence indicating that the PWWP domain interacts differently with the PWWP-truncated DNMT3A-DNMT3L complex in the presence of the nucleosome substrates?

Our structural analysis indicates that the PWWP-ADD interaction leads to occlusion of the H3K36me2-binding site in the PWWP domain. Therefore, in the active state, the interaction of the PWWP domain with the nucleosome substrate would conceivably compete against the ADD-PWWP interaction, leading to repositioning of the PWWP domain. Our unpublished data supports this notion. Following the Reviewer’s comments, we have clarified this point in the discussion section (Line 372-375) of the revised manuscript.

Response to Reviewer 2

Reviewer #2 (Remarks to the Author):

This is an exciting manuscript centered on DNMT3A/L structures that reveal a new autoregulatory mechanism, whose phenomenology has been described, but whose molecular details remained mysterious. The key breakthrough here is the new structural observation of a PWWP domain of DNMT3A packing against the enzyme active site and occluding histone recognition pockets, that is convincingly argued to be a novel autoinhibited state of the complex. The experiments are highly rigorous and combine the strengths of NMR and cryoEM, as well as convincing accompanying biochemistry to define the novel mode of PWWP interaction with the ADD and methyltransferase domains that serves to sharpen the previously noted nucleosome substrate specificity, that was thought to be purely driven by favorable binding enthalpy as opposed to a dynamic interplay of the PWWP domain occluding the active site and histone binding pockets. This goes beyond merely defining the structural and biochemical basis of H3K36me regulation of DNMT3a/L, the reanalysis of prior W330R mutant methylation patterns is consistent with the proposed model, but also can be interpreted as reduced affinity for

H3K36me3 by ablation of the aromatic cage that binds H3K36me3, a mutant that separates this function from PWWP autoinhibition would be needed to make a more robust conclusion from these data, but given the strength of the in vitro data, I am convinced that their newly discovered mechanism is operant in cells. Several noted disease associated mutations can be reinterpreted in light of the toggling in between the autoinhibited state and the open state, and these are tested in vitro as well. As this is an extremely important enzyme in cancer, development and epigenetic inheritance, this new regulatory mechanism will be of interest to a broad readership of Nature Communications, and likely extends to the related DNMT3B/L complex. While the gross features are similar to prior structures, one of the two 3A protomers in the tetramer has the PWWP domain modifiable, and it is oriented to block the DNA methyltransferase active site, the ADD domain that binds unmethylated H3 tail, and the PWWP's binding pocket that has been previously shown to bind H3K36me2. Evidence that the newly discovered autoinhibition interface is real and important is quite strong—mutations, ITC and NMR all support the interpretation that the PWWP-ADD autoinhibited interface observed in the new structure represents state in cells that is likely populated. Moreover, the inhibition of histone binding and methyltransferase activity being relieved by mutants that disrupt the PWWP-ADD and PWWP-MTase interfaces strengthens this interpretation. It is speculated based on TRD loop position and superposition with CpG DNA bound forms of the complex this PWWP packing inhibition mode would also occlude this interface, but it is not explicitly tested beyond and MD simulation (and would be challenging to actually test). Importantly, the authors also show that this autoinhibition mode also sharpens the nucleosomal substrate preferences previously noted in the literature for H3K4me0 and H3K36me2 for the ADD domain and PWWP-domain, respectively.

The Reviewer has summarized this study very well. We thank the Reviewer for his/her positive view of this work.

Major point—a mutant that disrupts the autoinhibited state that has clean separation of function from other individual module functions, tested in living cells would improve the case that this mechanism is relevant in cells, but I do not think is essential as the accumulated in vitro data is quite compelling on its own and consistent with prior cell and organism based studies.

We agree with the Reviewer that inclusion of a mutant that not only disrupts the autoinhibitory state but also has clear separation from other individual module functions in cells would further clarify the importance of this autoinhibitory mechanism in vivo. However, the lack of a detailed molecular understanding of the interaction between DNMT3A and its in vivo substrate, nucleosomes, makes a clear-cut design of such mutation rather challenging. To alleviate the Reviewer's concern, we have clarified this point by including the statement "Future investigation of such DNMT3A autoinhibition, e.g.

via a mutation that disrupts autoinhibition but not any other regulations of DNMT3A, may further clarify this link.” (Line 403-405) in the discussion in the revised manuscript.

Apart from a few typos, the paper is well written and figures well made to convey the key structural details, the rigor of the data is high and key points are made by multiple lines of unambiguous evidence.

We thank the reviewer again for his/her assessment of this paper.

Minor points:

Methyllysine analogs are used in this work rather to mimic the normal counterparts. Given noted deviation of MLAs from the native modifications observed in other contexts, I think this should be more prominently noted in the main text rather than appearing solely in the methods to avoid confusion.

Following the Reviewer’s suggestion, we have clarified this in the result section (Line 259-261).

The Kd reported for the ADD-PWWP interface needs to be appropriately truncated for significant digits (8+/-2 uM). Similar considerations apply to Fig 3b Kds.

We have rounded up the Kd values in Fig 2 & 3 accordingly.

A few typos to fix: line 56: “H3K36m2” , line 254 “bindings”

These typos have been fixed.

Response to Reviewer 3

In this manuscript the authors present a thorough and relevant study that nicely integrates structural information with biophysical, biochemical and molecular modeling studies to offer mechanistic insight into orchestration of the DNMT3A2/DNMT3L association with histone mark recognition, DNA substrate binding, and enzymatic methylation activity. This not only provides key mechanistic insight into the regulation of DNMT3A activity in normal cell function but importantly offers mechanistic perspective for how site-specific disease associated protein mutations disrupt key auto-inhibiting interactions to support observed hypermethylated DNA disease phenotypes. The inclusion of mutational studies in orthogonal biophysical and biochemical assays provides compelling support for the structural studies and the core conclusions presented. Overall, this is a comprehensive, timely and important investigation that advances our understanding of the molecular

mechanisms by which epigenetic alterations and aberrant DNA methylation accumulation may occur to promote disease conditions. The findings are highly valuable for advancing the field. Nevertheless, there are a few points that should be addressed prior to publication that would enhance clarity of presentation, strengthening the major findings.

We thank the Reviewer for his/her positive assessment of this work.

1) In the Abstract, most of the acronyms are defined, but not PWWP and ADD. It is recommended to define these to be consistent with defining all the other acronyms.

We thank the reviewer for the suggestion. We have defined the PWWP and ADD domains in the abstract in the revised manuscript.

2) Fig. 2: In Fig. 2d, as well as 3b, there does not seem to be error bars included on the presented ITC data, which should be added in.

We wish to clarify that the K_d values and error estimates in Fig. 2d & 3b were derived from two independent ITC experiments. For clarity, only one representative ITC curve was shown in these figures. Therefore, there are no error bars associated with the presented data. Following the reviewer's comment, we have clarified this in the figure legends in the revised manuscript.

Regarding the presented $^1\text{H}/^{15}\text{N}$ HSQC in Fig. 2e, it is difficult to differentiate the magenta from red peaks, making it difficult to readily evaluate the author's conclusions. It is also stated in the text that addition of WT PWWP results in "...substantial chemical shift perturbation of several..." ADD NMR resonances, though there seems to be very few ADD resonances that experience a chemical shift perturbation and the majority of the ones that do have small chemical shift perturbations. In addition, it is claimed that addition of the PWWP W330A variant shifts the ADD resonances "...back to.." the non-interacting form. The presented spectra support the conclusion that the ADD resonances move back toward the unbound ADD form, but do not revert back fully. It is thus recommended to attenuate the claims in these two statements. Having said that, it is again difficult to globally assess these claims given the lack of contrast between the red and magenta spectra.

We thank the reviewer for the suggestions. In the revised manuscript, we have toned down the statements for the NMR results, such as "the addition of the PWWP domain leads to notable chemical shift perturbation" (Line 183) and "replacement of WT PWWP with the W330A-mutated PWWP domain led to a shift of these peaks toward the positions" (Line 186) of the non-interacting form. In addition, we have changed the color for NMR signals associated with the PWWP W330A mutant from magenta to blue.

3) The Fig. 4 legend denotes f, rather than d. It is stated in regard to the DNA methylation assay data that neither variant R544A or W330A affected "...H3K4me0-mediated stimulation appreciably...". It is presumed this statement is being made relative to WT, in which it appears that the difference between H3K4me3 and H3K4me0 substrates in activating DNMT3A is greater for both variants than the 7-fold difference observed for the WT.

We apologize for the confusion. In the revised manuscript, we have clarified the description by stating that "the H3K4me0-mediated stimulation remains strong for both mutants (6.1-fold for R544A and 5.3-fold for W330A)" (Line 274-275).